# The Báa nnilah Program: Results of a Chronic-Illness Self-Management Cluster Randomized Trial with the Apsáalooke Nation

**DOI:** 10.3390/ijerph21030285

**Published:** 2024-02-29

**Authors:** Suzanne Held, Du Feng, Alma McCormick, Mark Schure, Lucille Other Medicine, John Hallett, Jillian Inouye, Sarah Allen, Shannon Holder, Brianna Bull Shows, Coleen Trottier, Alexi Kyro, Samantha Kropp, Nicole Turns Plenty

**Affiliations:** 1Department of Human Development & Community Health, Montana State University, Bozeman, MT 59717, USA; mark.schure@montana.edu (M.S.); shannon.holder@gallatin.mt.gov (S.H.); brianna.bullshows@gmail.com (B.B.S.); coleen.trottier@student.montana.edu (C.T.); lexi.kyro@gmail.com (A.K.); samantha.kropp@hsc.utah.edu (S.K.); 2Department of Nursing, University of Nevada, Las Vegas, NV 89154, USA; du.feng@unlv.edu; 3Messengers for Health, Crow Agency, MT 59022, USA; alma.mccormick@montana.edu (A.M.); l.othermedicine@montana.edu (L.O.M.); 4Petaluma Health Center, Petaluma, CA 94954, USA; johnhallett@phealthcenter.org; 5Manoa School of Nursing, University of Hawaii, Honolulu, HI 96822, USA; jinouye@hawaii.edu; 6Department of Family Life & Human Development, Southern Utah University, Cedar City, UT 84720, USA; sarahallen3@suu.edu; 7OneHealth Bighorn, Hardin, MT 59034, USA; nicole.turnsplenty@onechc.org

**Keywords:** United States, Indigenous, American Indians, chronic illness, community health, community-based participatory research, randomized controlled trial

## Abstract

Indigenous people in Montana are disproportionately affected by chronic illness (CI), a legacy of settler colonialism. Existing programs addressing CI self-management are not appropriate because they are not consonant with Indigenous cultures in general and the Apsáalooke culture specifically. A research partnership between the Apsáalooke (Crow Nation) non-profit organization Messengers for Health and Montana State University co-developed, implemented, and evaluated a CI self-management program for community members. This article examines qualitative and quantitative program impacts using a pragmatic cluster randomized clinical trial design with intervention and waitlist control arms. The quantitative and qualitative data resulted in different stories on the impact of the Báa nnilah program. Neither of the quantitative hypotheses were supported with one exception. The qualitative data showed substantial positive outcomes across multiple areas. We examine why the data sets led to two very different stories, and provide study strengths and limitations, recommendations, and future directions.

## 1. Introduction

The Báa nnilah program was designed to support Apsáalooke (Crow) Nation community members in the self-management of their chronic illness (CI) and to improve their overall well-being. Existing programs that seek to lower mortality and increase quality of life through CI self-management [1,2] are not appropriate because they are not consonant with Indigenous cultures in general and the Apsáalooke culture specifically. Báa nnilah was developed, implemented, and evaluated in a partnership between Messengers for Health, an Apsáalooke non-profit organization, and Montana State University. The partnership began in 1996 and uses Indigenous research and community-based participatory research approaches to address community-identified health topics using cultural strengths and the expertise of community members.

This paper reports the qualitative and quantitative outcomes of the program. Other published papers discuss the need for this program from the standpoint of the community and from population health data, as well as providing information on the development of the program, the program’s content and approach, and details of the randomized clinical trial protocol [3,4,5,6]. 

The Báa nnilah (meaning to give advice) program was designed based on a collaborative story analysis of interviews with Apsáalooke community members living with a CI (for more details, see [7]). Apsáalooke spirituality and language are both foundational and embedded features of the Báa nnilah program. As a CI self-management program rooted in Apsáalooke strengths and resilience, Báa nnilah (1) had community mentors with CIs (Aaakbaabaaniilea, ones who give advice) to recruit participants and facilitate program gatherings, (2) had participants learn through advice given by other participants and their Aaakbaabaaniilea and via personal and collective stories, and (3) nurtured a safe and supportive learning environment. The program was organized into seven gatherings (titles followed by content areas): (1) Beginning the Báa nnilah journey (program introduction), (2) Ongoing conditions and self-care (understanding chronic illnesses), (3) Daasachchuchik—Strong Heart (understanding historical and current trauma and the role of resilience), (4) Healthy food and physical activity (learning about healthy eating and physical activity), (5) Positive healthcare experiences (learning how to develop and maximize positive healthcare experiences), (6) Healthy communication and overcoming challenges (learning how to be a self-advocate for one’s self), and (7) Closing and graduation (celebrating and reflecting on one’s experience with the program). 

## 2. Materials and Methods

This pragmatic clinical trial consisted of an intervention arm and a waitlist control (WLC) arm. We hypothesized that intervention participants would have significant improvements in their quality of life compared to WLC participants, immediately following the intervention, and at 6 and 12 months post-intervention. We also hypothesized that, compared to the WLC participants, intervention participants would show significant improvements in (a) measures of physical function, (b) self-efficacy for managing CI symptoms, (c) depression, (d) patient activation, (e) emotional support, (f) self-efficacy for managing emotions and social interactions, (g) positive affect and well-being, and (h) satisfaction with participation in social roles. We used family/clan group randomization, as individual-level randomization raises implementation challenges because (1) it is impractical to have individual family members receive health-related interventions, while preventing other family members from receiving those interventions; (2) it is culturally inconsonant to require that family or friends not share what they learned that could result in better health and a higher quality of life; and (3) outcomes within family/clan clusters are likely to be related. Thus, we used a cluster randomized trial (CRT) design. To maximize the number of clusters, each cluster was a group of participants belonging to the same family/clan. Maximizing the number of randomization units in CRT increases statistical power and reduces potential bias due to an imbalance of baseline cluster characteristics [8]. This study was approved by the Montana State University Institutional Review Board (IRB). There was not an active IRB on the Apsáalooke Reservation at the time of this study; we received a letter of support from the tribal executive branch. 

Recruitment, enrollment, and randomization: The inclusion criteria included (a) age ≥ 25 years, (b) being Indigenous, (c) living on or near the Apsáalooke Reservation, and (d) being diagnosed with at least one CI. The exclusion criterion was having an advanced terminal condition. Aaakbaabaaniilea screened for study eligibility and recruited members in their community, prioritizing family members. For details on study recruitment, retention, and adherence see [9].

Each Aaakbaabaaniilea (*N* = 10) enrolled approximately 20 participants (N = 211) and identified which participants belonged to the same family/clan. Each participant and each family/clan cluster were assigned a unique ID. The clusters enrolled by the same Aaakbaabaaniilea were sorted by size, ranging from 1 to 8 participants per cluster, and then randomly assigned to the intervention (*n* = 104) or WLC group (*n* = 107). Our power analysis revealed that the proposed sample size would achieve 80% power for detecting overall intervention condition differences in the primary outcome and at least 80% power for the secondary outcomes, with a conservatively estimated 0.6 correlation between repeated measures, an intraclass correlation (ICC) of 0.1, and a 20% attrition rate. This sample size would also provide 80% power to detect a small (f = 0.1) intervention-by-time interaction in an outcome. 

### 2.1. Quantitative Data Collection

Quantitative study outcomes were measured at four time points (i.e., baseline [T1], post-intervention [T2], 6-month follow-up [T3], and 12-month follow-up [T4]) among the intervention group, and a fifth time for the WLC group (i.e., pre-baseline [T1], baseline [T2], post-intervention [T3], 6-month follow-up [T4], and 12-month follow-up [T5]). Survey measures were administered with computer tablets via Qualtrics. Physical tests were monitored and recorded by research team members. Assessments were completed during scheduled meetings in the communities of study participants. For participants unable to attend scheduled data collection meetings, make-up assessments were conducted. Among the 104 participants in the intervention group who provided baseline data, 83 (80%), 72 (69%), and 69 (66%) participants participated at post-intervention, 6-month follow-up, and 12-month follow-up, respectively. Among the 107 participants in the WLC group who provided pre-baseline data, 90 (84%), 75 (70%), 73 (68%), and 74 (69%) participants returned for data collection at baseline, post-intervention, 6-month follow-up, and 12-month follow-up, respectively. Participants were contacted to ensure accuracy of self-report data for outlier responses.

### 2.2. Quantitative Data Measures

We utilized validated health surveys and physical tests to assess the primary outcome of quality of life and secondary outcomes, including social/emotional function, depression, physical health, and patient activation. Báa nnilah project staff evaluated measures for appropriateness and made changes where necessary, detailed by Hallett et al. [6].

Short Form Health Survey-12 (SF-12), Physical and Mental (version 1): SF-12 measures quality of life and was considered culturally appropriate for Apsáalooke participants after we received approval to modify moderate physical activities by deleting golf and bowling and adding carrying a small child, walking for exercise, and round dancing/push dancing. The split-half reliability (Pearson’s r between the 2 items per component) was 0.72 for physical functioning, 0.71 for physical role, 0.43 for emotional role, and 0.37 for mental health. SF-12 summary scores for physical wellbeing (PCS) and mental wellbeing (MCS) were computed based on a correlated (oblique) factor solution recommended in the existing literature [10].

Patient Health Questionnaire-9: PHQ-9 measures the degree of depression symptoms over the last two weeks and has been previously validated for an Indigenous cohort [11,12]. The Cronbach alpha of the PHQ-9 was 0.92.

Patient-Reported Outcome Measurement Information System: PROMIS includes self-reported measures of physical, mental, and social dimensions of health, which have been validated across many CIs [8,13,14,15,16]. We utilized the following measures: Satisfaction with Social Roles and Activities [17,18], Self-Efficacy for Managing Symptoms [19], Emotional Support [20], Positive Affect and Well-Being [21], Physical Function [8,19], Self-Efficacy for Managing Emotions [19,22], and Self-Efficacy for Managing Social Interactions [19]. The Cronbach alpha for these measures ranged from 0.91 to 0.98.

Patient Activation Measure: the PAM assesses self-management-related health knowledge, skills, and confidence [23,24]. We were unable to calculate reliability due to the proprietary scoring scheme.

Modified Timed Up and Go: mTUG measures mobility and balance deficits, which correlate with increased fall risks [25]. Participants rise from a seated position, walk 3 m, return, and sit. For this study, the distance walked was modified from 3 m to 10 m due to an error from a consultant who relayed the incorrect distance for this measure to our research group.

Modified Balance Error Scoring System: The mBESS assesses static balance [26,27]. Errors are counted as participants stand for 20 s with their eyes closed (1) on both feet, (2) on one foot, and (3) with one foot in front of the other. Lower mBESS scores indicate better balance. Changes for our data collection include testing single-leg stance with the dominant foot as opposed to non-dominant foot and allowing participants to determine their lead foot during the third stance. We decided to alter the test to make it more possible for participants to complete as they had chronic health conditions and this test proved difficult for them when we piloted it.

Six-minute walk: This serves as a corollary for physical function and cardiovascular performance [28,29,30]. Participants walked at a comfortable pace for six minutes and their distance was recorded.

Statistical analysis: Statistical analyses were conducted using SAS 9.4^®^ [31]. Descriptive statistics were obtained for all study variables. The mean and standard deviation were calculated for continuous variables, and frequency and percentage for categorical variables. Distributions of continuous variables were examined for non-normality and outliers. Missing data patterns were examined for all variables. To compare the two study groups on baseline demographic characteristics and outcome measures, independent t-tests were conducted for the continuous variables, and chi-square tests for the categorical variables.

Observations were nested within individuals and individuals nested within family/clan clusters. Up to four observations (i.e., measurements over four waves of data collection) per participant in the intervention group (*n* = 104, number of clusters = 45) and up to five observations per participant in the WLC group (*n* = 107, number of clusters = 50) were obtained. The theoretical expectations were that participants in both groups would show improvements on the outcome measures post-intervention, and that the intervention effects would be sustained at follow-ups. A hierarchical linear modeling (HLM) approach was adopted, and each main hypothesis was tested in SAS 9.4 using the Proc Mixed procedure. A dummy-coded time-varying covariate (“intervention”) was created to indicate whether each observation was pre- (coded “0”) or post-intervention (coded “1”). The significance of the intervention effect is tested by this intervention variable. Time-invariant covariates included age at baseline, sex, study group membership, number of chronic illnesses, and number of gatherings attended. A Time (i.e., number of weeks since baseline) by Group interaction effect was also included, because if the intervention effects were significant and sustained through the follow-up period, the intervention group would have better overall outcomes compared to the WLC group. The number of valid data points varied across individuals, as well as by time and by variable within each individual, because participants could (1) miss a data collection but complete subsequent data collection(s) and/or (2) choose not to complete a physical test or a survey question. Full information maximum likelihood estimation (MLE) was used to handle missing data, which allowed us to use all available non-missing data.

### 2.3. Qualitative Data Collection

Three sets of qualitative data were used to assess changes in participants and the community due to the intervention. The first set of data were from semi-structured interviews with participants. We selected a random sample of 20 participants (for a 10% random sample) and the community Co-Principal Investigator conducted 13 interviews at a location and time convenient for participants within two months after program completion. Some participants were not able to complete interviews due to personal family obligations and family death. The sample included 11 females and 2 males with an age range of 24 to 72 and a mean age of 54.2, from different communities with different Aaakbaabaaniilea, across both the intervention and wait-list control groups and with a variety of CIs. Project partners developed a semi-structured interview guide to learn about impacts of the program on participants and the broader community (see [6] for interview guide). The interview started with a broad question about the influence of the program on the participant and included questions on the most and least helpful aspects of the program and suggestions on what could be changed to make the program better. Interviews were conducted in English and lasted from 12 min 10 s to 57 min 51 s, with an average of 25 min 12 s. A transcription guide was developed, and interviews were transcribed verbatim using semi-naturalized transcription [32,33]. We refer to this data set in the results as “interview data”.

The second set of data were information entered into tablet computers by participants. The question “Please share what you gained or learned from the program and what changes in your health or behavior you made because of the program.” was the last question on the tablet computers for the last two data collection time points. Participants could type in a response or leave it blank. At the fourth time point, 104 out of 143 participants (73%) entered comments into the tablet computers, and at the fifth time point, 69 out of 74 (93%) entered comments. For the fifth time point, data were gathered only for the waitlist control group. We refer to this data set in the results as “tablet data”.

The third set of data were informal information shared by participants with program staff and Aaakbaabaaniilea. These data were oral, conversational, and storytelling-based, and took place within the context of a trusted relationship, allowing participants to use their own words. Participants shared program impacts during the data collection gatherings or during casual encounters in the community. These encounters happen frequently, and community members felt free to share due to living in a small rural community, the close-knit nature of the Apsáalooke, and the many social activities and celebrations that occur. We refer to this data set in the results as “story data”.

### 2.4. Qualitative Data Analysis

For the interview data, we used a variation of a qualitative data analysis method previously developed by this team. This method aligns with Apsáalooke culture and honors and respects the participant by keeping their story whole rather than breaking it apart into themes (see Hallett et al. for details on this method) [7]. To analyze the interviews, a sub-group (Indigenous and non-Indigenous) of the overall partnership team read interview transcripts and held multiple analysis sessions, where we discussed “what touched our heart” from each interview. During these sessions, a conceptual model to describe program impacts naturally emerged. Team members visualized the image of a teepee, the traditional ancestral home of the Apsáalooke (Ashtáale—your real home) to appropriately illustrate the impact of the Báa nnilah program. This conceptual model is deeply rooted in Apsáalooke values, symbolism, and beliefs. For the interview data, for each participant, we created a story summary highlighting how impacts of the program are represented by elements of the teepee model. The story allows the reader to hear the voice of the participant and to feel the power of the story and allow it to become alive. For the tablet and story data sets, illustrative quotes were woven into the findings, supporting the conceptual model. Recognizing the interconnected nature of how impacts of the program were shared, we present findings from the tablet and story data centering the most salient component of Ashtáale with notes of how their story also reflected other model components. Looking at data from these three sources provides multiple perspectives of the program effects, like the view from a Crow’s nest.

## 3. Results

### 3.1. Quantitative Data Results

There were 135 of 153 participants with complete attendance data, and of those, the majority (55%) attended all seven intervention gatherings and 79% attended at least five gatherings. Table 1 shows the demographic, background, and health characteristics of the study sample (*N* = 211). As seen in Table 1, approximately 33% of the participants had an annual household income less than USD 10,000 and another one-third (34%) had an annual household income between USD 10,000 and USD 24,999, whereas only 8% had an annual household income of USD 50,000 or greater. All participants had at least one chronic illness, with the majority (70%) having comorbidities, and 40% having three or more comorbidities. The most common chronic illnesses were diabetes (57%), high blood pressure (56%), chronic pain (35%), and arthritis (34%). Among participants who had diabetes, 96% of them had type II diabetes. On average, the participants had approximately four doctor visits in the past four months (mean = 4.13, SD = 6.35) and traveled about 19 miles (mean = 18.91 miles, SD = 19.63 miles) one-way to reach their primary care clinic. The one-way distance to the clinic ranged from less than a mile (n = 28, 14%) to 50+ miles (*n* = 17, 9%), with the longest distance being 90 miles. More participants in the intervention group had diabetes (65%) compared with the WLC group (49%); *p* = 0.014. Also, more participants in the intervention group had private insurance (16%) compared with the WLC group (6%); *p* = 0.012. No other significant demographic differences were found between the two groups.

Table 2 shows the baseline mean and standard deviation (SD) of the three physical tests and 11 health survey outcome measures by study group for T1 and T2. Independent t-tests revealed that, on average, participants in the intervention group scored lower on the SF-12 Physical Wellbeing (*p* = 0.034) and on PHQ-9 for depression (*p* = 0.040) compared with the WLC group. No other significant differences were found when comparing group means.

Table 3 presents the results of the HLM analyses to test longitudinal trends by group and intervention effects. As mentioned above, the intervention effect is tested by the dummy-coded time-varying covariate named “intervention.” Except for the TUG Mean and mBESS, there were no observed intervention effects. There was a significant Time by Group interaction effect on the TUG Mean (B = −0.006, *p* < 0.001) and mBESS total (B = 0.009, *p* < 0.01) indicating significant group differences in changes in these outcome measures, in that the intervention group had a greater decrease in time for TUG Mean (i.e., greater improvement in mobility and deficits in balance), but a higher number of errors in the mBESS (i.e., greater decline in static balance), compared with the WLC group, controlling for all other covariates.

As seen in Table 3, there was an observed significant time effect on mBESS total (B = 0.01, *p* < 0.001), the SF-12 Physical Wellbeing subscale (SF-12 PCS) score (B = 0.01, *p* < 0.05), the SF-12 Mental Wellbeing subscale (SF-12 MCS) score (B = 0.01, *p* < 0.01) and the PAM (B = 0.01, *p* < 0.05), indicating that the mean scores for these measures significantly improved over time regardless of group assignment. Among the time-invariant covariates, the positive significant sex (1 = male; 0 = female) effect on PROMIS Physical Functioning (B = 2.91, *p* < 0.05) indicates that, taking into consideration all other effects in the HLM model, men scored better on physical functioning than women across all data collection time points. Age had a significant effect on TUG Mean (B = 0.17, *p* < 0.001), mBESS total (B = 0.19, *p* < 0.001), SF-12 Physical Wellbeing (B = −0.10, *p* < 0.05), and PROMIS Physical Functioning (B = −0.21, *p* < 0.001), indicating that younger participants scored better on these measures than older participants. The number of CIs had a significant effect on SF-12 Physical Wellbeing (B = −2.70, *p* < 0.001), PROMIS Physical Functioning (B = −2.49, *p* < 0.001), Self-Efficacy for Managing Symptoms (B = −1.43, *p* < 0.01), and Satisfaction with Social Roles and Activities (B = −1.77, *p* < 0.01), where participants who reported fewer CIs scored better on these measures. Educational level had a significant effect on the PAM, where participants who had a high school diploma/GED (B = 9.07, *p* < 0.05) and those who graduated from college or above (B = 10.28, *p* < 0.05) scored better compared with participants whose formal schooling was up to the 8th grade.

### 3.2. Qualitative Data Results

For the Apsáalooke community, the terms teepee, home, and lodge are interchangeable. The Apsáalooke-style Ashtáale includes four base poles, canvas, stakes, and the base. Ashtáale is considered the Apsáalooke people’s second mother and is a sacred place. She provides shelter and protection and is where community members feel safe to share their voices. It is where stories are shared, lessons are taught, and change occurs. A good home is where “singing and cheery voices” are heard, even when the weather is inclement [34]. At the heart of the Ashtáale model is connection and relationality among all pieces, with each piece performing a vital function to make the home for the Báa nnilah program complete, stable, and unified. Figure 1 displays the conceptual model for the qualitative data results and Box 1 provides participant stories of the impact of the program.

The four base poles of Ashtáale represent the impact of the program on (1) the participant, (2) their household, (3) their family, and (4) the broader community. The first pole describes the relational change(s) an individual experiences with their current and future health behaviors, attitudes, and beliefs due to the influence of the program. Individual changes in the self for the self are situated within relational purpose, such as making changes so one can be there for others and be a role model.

Sample quotes:


*“I learned to take my health concerns more seriously and to ask questions of my health care providers when I have medical appointments. I feel more confident in improving my health”*
(tablet data).


*“Learned I gained friendships, trust in those that we share some same experiences. It takes a community to learn & teach from each other”*
(tablet data).

One participant shared a story to staff members (story data) that she was diagnosed with Hepatitis C acquired from sharing needles during a time she was struggling with addiction. She felt embarrassed about this diagnosis and lived in fear of spreading the disease to others, causing her to not seek treatment for the disease, avoid healthcare providers, and feel bound up in shame about her past behavior. She shared that this program enabled her to release the shame and gave her courage to go in for treatment. She explained, through tears of happiness, that she no longer had Hepatitis C. Her next goal is to share her story with community members who had similar experiences to let them know that they could be healed from this disease and live an emotionally, physically, and spiritually healthier life (also reflecting poles 2, 3, and 4).

Another participant shared with staff (story data) during a data collection event that, due to the program, he lost 50 pounds, and that many of his healthcare problems had resolved. Together, his family applied what they learned through Báa nnilah and started walking, eating healthier, and staying away from sugar-sweetened drinks (also reflecting poles 2 and 3).

A mentor shared (story data) that upon receiving the medication log that was provided as an incentive gift, a participant immediately started filling it out with information on her medications and felt empowered and had improved confidence to know more about them. The medication log is a piece of laminated 8 ½ × 11 paper with spaces to write the name and dose of different medications, what they are for, when to take them, what they look like, and when a refill is needed. Another mentor shared (story data) that other community members heard about the usefulness of the medication log, how it improved their management of the CI, and asked for them.

An Aaakbaabaaniilea shared that they applied information they learned from facilitating the program and were able to stop taking some of their CI medications. The doctor said to him, “I don’t know what you are doing, but you don’t need to take these medications anymore”. The Aaakbaabaaniilea went on to say, “I’m in it to heal and help others to heal and that’s the beauty of this program” (story data; also reflecting pole 4).

A participant shared that after her grandfather whom she was very close to passed away, she was devastated with grief and became very depressed (story data). She quit taking medications for her CI and did not attend any medical appointments for about two years. She was so depressed she considered suicide and had made plans to do so. She expressed that her participation in this program “brought me out of my depression” and due to the program’s impact, did not follow through with her plans. Currently, she is still doing very well.

The second pole describes relational change(s) experienced by family members living in the home with a program participant due to the influence of the program. This includes multi-directional influences, for example, from the participant to a household member and back to the participant, or from the participant to the household member to other household members. For the Apsáalooke, it is natural for households to be fluid and to hold three or four generations, as family members care for each other, especially elders, by welcoming them to stay in their home for short or longer stays. Currently, a lack of housing and poverty also play a role in the fluid nature of households.

A couple who took part in the program together shared with program staff (story data) that in addition to learning how to improve their communication with healthcare providers, the program taught them how to better communicate and understand each other and have a stronger relationship. They are now able to express how they are feeling and are more patient with each other. These quotes are from the tablet data:


*“The program has opened my eyes towards my health along with my children’s health.”*



*“To take care of your family you need to take care of yourself first.”*



*“I gained a better sense of partnership with my wife on health issues.”*


The third pole describes relational change(s) that occur at the family level in the participants’ current and future health behaviors, attitudes, and beliefs due to the influence of the program. Among the Apsáalooke, when someone receives something positive, it is natural and appropriate to share it with others. Apsáalooke are close-knit, with family ties and the clan system being extremely important. These quotes are from tablet data:


*“I became more active with my family.”*



*“Improved eating and drinking habits, able to share this with extended family.”*



*“I’ve had a good experience with the program. It’s good to know that I’m not the only one that is [going] through the things I do. I was also able to share with my adult children the things I learned about managing my health. My two oldest sons have taken control of their weight issues and have lost weight. They are no longer on high blood pressure medication or diabetes medication! I hope to do the same.”*


A participant shared with an Aaakbaabaaniilea (story data) that they had low self-esteem from being diabetic and that they felt like a burden on their family and didn’t talk about their illness with them. Through the program, they gained a better understanding of their illness and were able to express themselves to their family members, which helped them to feel better and to take action on behalf of their health.

Multiple participants shared (story data) that they read through the information in the program manual describing various CIs to gain knowledge to help their relatives. For example, a participant who did not have diabetes would read about diabetes in the program manual and share that information with family members who had that CI.

One of the participants was very sick and told her Aaakbaabaaniilea (story data) that she knew she could not save herself, but that she used information from the program to save her children, her sisters, and her other family members. The participant passed away and the family members share with the Aaakbaabaaniilea that they continue to use the manual and information as a way to honor and reminisce about their mother while helping themselves to be healthy.

The fourth pole describes relational change(s) that occurred at the community level in current and future health behaviors, attitudes, and beliefs due to the influence of the program. The cultural strength of sharing described above occurs at the family and broader community level. These quotes are from tablet data:


*“I gained relationships with new people and created deeper bonds with people who are already in my life. I was able to hear the stories of other people who share my condition and feelings and learn that I’m not alone as I navigate my health issues.”*



*“I have a support outside the doc[tor]. I have made new friends and people in health.”*


The next three components of the Ashtáale (base, canvas, stakes) are what Aaakbaabaaniilea and Báa nnilah program staff (story data) concluded were necessary to generate the impacts discussed above under the four poles. The first component is the base, which is the foundation of the Ashtáale, and in this model, represents the foundations and roots of the Báa nnilah program. This includes the cultural strengths of stories, spending time together connecting and sharing a meal, the Apsáalooke language, spirituality, kinship ties and the clan system, and the use of traditional stories and Apsáalooke traditional designs and colors in program material. Cultural strengths, ground rules established in the first gathering (hope, acceptance, respect, trust, plus others added by each group), and a confidentiality agreement signed by all Aaakbaabaaniilea and participants assured an atmosphere where community members could openly share, develop deep connections, feel safe and comfortable, and give and receive support. For more information on the cultural components of the program, see Held et al. and Schure et al. [3,4].

The second element necessary for impacts is the canvas, which covers the Ashtáale, and is represented by Messengers for Health’s Executive Director and community Co-Principal Investigator, Alma Knows His Gun McCormick. She is a respected elder who is from the community, speaks her language, is a leader, and is seen as someone with integrity. People know they can reach out to her for assistance, guidance, and prayer. She has had hardships and community members observed how she dealt with them, making her a role model for others. She holds herself accountable to her community. Báa nnilah would not have happened without her as she created the atmosphere to bring people together and provided trust, safety, shelter, hope, and love. Her leadership allowed community members to take steps toward healing. Her love for her people held everyone together, covering and protecting them like the canvas protects those in the Ashtáale. Her Apsáalooke name, Sacred Shield, portrays this as well and reflects prayers passed down through generations for her to become the woman she is today. As one participant expressed (story data), “*If Alma’s a part of this, it must be good*”.

The final element necessary for impacts are Ashtáale’s stakes, which anchor Ashtáale to the ground. In the model, the stakes represent the continuity, stability, and sustainability of the program. This refers to both the trust and integrity that Messengers for Health built in the community prior to the Báa nnilah program and also the hope for program sustainability. Regarding trust and integrity, one participant shared (story data), “*I know I can always call upon Messengers for Health if I ever need help. I am a healthier person for all the help and information they have shared with all of us. Definitely an oasis in our community!!!! Ahó [Thank you]*.” Sustainability includes participants becoming mentors to other community members, as one participant stated (story data), “*This program has been invaluable to me! Perfect time in my life. I hope to be a mentor one day*.” Representative quotes from participants (story data) hoping for sustainability are “*I wish that more people would give it a chance. I wish that there was more room for more people*.” and “*Would like to have this program continue to help others that need it more*”.

Box 1Participant stories.                       **Prays Good’s story**Prays Good provides us with strong examples of the teepee model in action and specifically the stakes and all four poles. Prays Good offers examples of changes she made for herself and a story of transformation in her relationship with her healthcare provider (Pole 1, relational changes with self): …there was some stuff they had us do, exercises they had us do…I started doing that…and now, I can walk up the steps…I can go up the stairs by myself without any of my grandchildren holding me. When I first went in there, she, she just puts her head in her computer. And she turns around. ‘Do you need something else?’ I tell her no and that’s all I say. That’s all I said. And then, ‘Alright your meds will be at the pharmacy.’ And I walk out, and I go take my meds. But now I started talking to her and…as soon as I talk more and then she started talking…she lets her computer go and she turns around towards me and we start talking…And so, we understand each other more now. Another example of the teepee model is her role as a Kaale (grandmother) and the reciprocity of these relationships (poles 2 and 3, relational changes in household and family):  I got my grandsons to do it with me…One day I sat with them and I was reading the book [the program manual] and I said ‘we need to change. You’re young,’ I said, ‘even young kids can be diabetic.’ And so, the two older ones quit pop. And they kept saying, Kaale, ‘You need to quit pop. You’re drinking too much pop.’
 The conversation with her grandchildren inspired her to change what she was drinking. She shares a story about her joy in learning that what she imparted to her grandsons was applied and taken to heart. Her grandsons went to McDonalds and made healthy choices by ordering salads “without me there with them, they did it on their own.” Prays Good takes on the role of an Aaakbaabaaniilea and shares what she learned with other community members (stakes, program continuity and sustainability): So, I’m always telling people about it. I had my book [the program manual] one time and we were going to a birthday party and…that was about all I was talking about at the birthday party. She recalls a story of going to her neighbor’s house (stakes, program continuity and sustainability; pole 4, relational changes in community) and “tell[ing] them about my meetings and the stuff that we’re talking about at the meeting.” Her neighbor shared that he was borderline diabetic and asked her, “Well, what can I do to get off [medications]?” She provided information to him on food portion sizes, how to cook, what to eat and the importance of exercise. He applied her advice, “and he started it, so now he’s not diabetic.” She shared another story of giving advice and encouraging a young woman who was struggling with alcohol use who applied her guidance and went to treatment, changed her friends, and her life. She encouraged the young woman to take on the role of an Aaakbaabaaniilea, “I said, you can tell them [others who are using alcohol] what I told you. You can use that and try to get them out too.”                        **Sacred Medicine’s story**Sacred Medicine provides many examples of the ways in which the Báa nnilah program impacted herself, her family, and community. She begins by sharing changes she noticed within herself (pole 1, relational changes with self): It really helped me know what I was eating and what I should do to take care of myself…I lost a lot of weight because I don’t eat a lot of the junk food I did before… And I’m getting into some of the vegetables…fruits too… and I’m walking every day now. She also explains how the program improved her self-management and her relationship and communication with her doctor (pole 1, relational changes with self): I have been making all my appointments…I tell her [the doctor] more… [before] I would keep everything in. She would ask me “Are you okay?” and I would say “Yeah”. But now I’ve been telling her what I’ve been going through and tell her what’s really going on instead of saying I’m okay. The program also inspired her to became more open and honest about her health with community members (pole 1, relational changes with self). She mentions how in the past, “I barely talk about my health with anybody and ya know ‘I’m okay’ and now I’m sharing what I’m going through and what they are.” She “realize[s] that I need to take care of myself since my grandkids are there… you have to be there for them as long as you can…[and] teach my grandkids into health…and just watch over my family.” This realization “that I *can* take care of myself” (emphasis by participant), is helping her to also take care of others, which is resulting in members of her family making healthy choices. She takes on the role of being a mentor, impacting her household and family (stakes, program continuity and sustainability; poles 2 and 3, relational changes in household and family): I’m always talking to my daughters and my grandkids…my two grandsons…I get after them, like check that, check that. They drink a lot of pop, so I get after them then I go and tell them the labels and stuff. And here now they quit drinking pop and they’re drinking more tea and water and flavored water… they started running…and my older one, he’s 16, and was really chubby before they started all of that. He musta weighed like 228 or something… he’s now 180 I think so he really likes that he lost weight…And the other one, he lost more. And my daughter too—she’s diabetic, and so I talked to her about it and had her read some stuff and… my mom just like four years ago she was pre-diabetic and she’s doing pretty good and starts to walk… I’ve gotta push to get her to eat healthy… She used to take the shortening and lard and now every time she does that I kinda get after her. But now she uses vegetable oil…and she’s been good with her diabetes and my daughter really watches herself.  These changes in her household and family had a broader impact on her community (pole 4, relational changes in community). She voices what happens when her grandson comes to play basketball: The other day he came and there’s a bunch of kids running around all over, which I thought was good, because you don’t usually see, usually there’s no kids … they’re always sitting in front of the TV. She recollects how her Aaakbaabaaniilea centered Apsáalooke cultural strengths such as spirituality, sharing, stories, and creating a safe and comfortable place (base, cultural roots of the Báa nnilah program): She would open up with a prayer…and she would explain everything and then she would have us talk about what we thought about everything, so including everybody. I thought it was great because we each had our own, ya know, what they were going through.                          **Sacred Cedar’s Story**Sacred Cedar provides evidence of the holistic impact of the program on her life and how she applied what she learned with her family. She begins by stating how the program impacts her (pole 1, relational changes with self): The program…focuses on living a healthier lifestyle. So that really made me think about how I needed to look at the things that I eat and focusing more on having a better attitude with my environment. And making better choices with my eating habits. Overall to take a good look at the things that I need to work on in my own self… I make healthier choices when I eat out. Before, I always treated myself [laughter]. I am making healthier choices by really looking at what I can have on the menu. So those are some of the things that I am working on.  Communicating better with your doctor…one of the things that our mentor shared was to write things down. So that was a good one for me. I used that. So now I tend to write it down like what were the things that were concerning me that I need to ask…and when I’m there then I’ll remember them. So that’s been one of the helpful things that I’ve done that I’ve used. Sacred Cedar recalls the influence of her mentor and the sharing that happened among participants (poles 1 and 4, relational changes with self and in community; base, cultural roots of the Báa nnilah program; stakes, program continuity and sustainability):   She [her mentor] imparted information like healthier lifestyle by exercising more, creating a healthier atmosphere for your mind and even in your heart by sharing things that you want to work on to improve yourself. The sharing really made me look at the things that I need to work on to be healthy, my mind, body, and my soul. …I think at first, I got offended because I was a jerk. And I didn’t want to share things. She [her mentor] would encourage everyone to share something that is bothering them or…anything improving in your life or things that are important to you. So that other participants would be able to have a different outlook on that too.  I talked to a co-worker about it [historical trauma] and how…when we have historical trauma in our life and we tend not to deal with it, it hinders us in our daily walk in life. But when we talk about that historical trauma and we share it, that’s just sharing one aspect of what life is about. Because as Native people, we all have a historical trauma in our lives at some point. And to just know that another person is going through the same thing. And how they dealt with it, those really important points. I shared a lot of information about that, like cultural and healthier lifestyles, eating healthier and exercising. Sacred Cedar transforms what she learned in the program into action on behalf of the health of her family and others (poles 1 and 3, relational changes with self and in family; stakes, program continuity and sustainability): I used to do my own little healthy thing. But then with this Báa nnilah program, that really made me look at what I actually have to do for long term. Not a quick thing here and then just leave it. So, the long-term of looking at your health is what really helped me to understand and so that’s one of the things that I try to talk to people about is that long term health of things that we can do to improve on this, improve on our healthy living like life, eating, just the whole thing, you know, holistic, holistic as we are as people. Before I didn’t really talk about my health with anybody. I talk about other things. But now I tend to bring up healthier eating habits, even to my own family. And I have siblings and some of them are diabetic and so I would talk to them about eating healthier and staying away from soda. Because soda is such a big part of our life. And giving that up for some people is so difficult. So, I’m always talking about eating healthier and encouraging more of my family members to start walking and trying to do some exercises and making that time to do that.                         **Earns with his Rope’s story**Earns with his Rope provides us with examples of learning through Apsáalooke cultural strengths and fulfilling his role as his grandparent’s eldest grandchild. He shares changes he made to be “healthy, happy and keeping active” (pole 1, relational changes with self): I familiarized myself with ways to help with my self-care, that program brought it out more in us. It brought out more for me to exercise, to move around. And for my age group, it is important for the older generation to help themselves by exercising and dealing with their disease. I am always moving, trying to get fit, lifting weights, running…I was given the band to exercise with, which is useful. Even while sitting, I can exercise.  I learned more about different diseases and… I learned more in eating habits. People think it’s okay to eat junk food, but too much sugar in many items like soft drinks are not good for you. I hardly drink any soft drinks now. I drink more water, healthy drinks…I became more aware through the program to take better care of myself, be aware how to take care of myself…The topics broadened my horizon. Chronic illness is an important topic…before I didn’t know much about it. Now I know.  The participant shares the impacts of learning through Apsáalooke cultural strengths, including storytelling, humor, and sharing what touches your heart with others (base, cultural roots of the Báa nnilah program; stakes, program continuity and sustainability):  I learned through the storytelling. It is really important to tell stories …What I learned is that people sharing stories help me to learn. Laughter is good. I learned useful information. There was good interaction. What I learned I keep in my heart. Being healthy is important…. If we’re at a meeting or get together, we need to share how we eat, drink less sugar, drink more water and in general watch what we eat…I knew this before but sometimes I forget to share. Earns with his Rope shares about the role he has in his family and how he is fulfilling this role. It is appropriate in the Apsáalooke culture for grandparents to raise their oldest grandchild. Grandmothers’ grandchildren are known to receive much advice and are expected to take on the role of imparting words of advice, gifts, and prayer to others (stakes, program continuity and sustainability; poles 2 and 3, relational changes in household and family).  I am my grandma’s oldest grandchild, so I talk to my family in eating what they should eat and drinking what they should drink. Sometimes if they are eating something that is not good for them, I will remind them to eat something healthier and if they are drinking pop or something sugary, I will advise them to drink more water.  Earns with his Rope shares how he believes Báa nnilah could have a broad positive impact and be sustained in the community (stakes, program continuity and sustainability): Just make it more available to everybody else. More people would know, working with other people or for everybody.

## 4. Discussion

The quantitative and qualitative data tell different stories of the impact of the Báa nnilah program. The main hypotheses were not supported by the quantitative data, given that no significant intervention effect was found (see the column labeled “Intervention” in Table 3), and only TUG Mean showed a significant Time by Group effect, favoring the intervention group. However, the story of outcomes from the qualitative data is one of substantial impact for participants, their families, and the broader community across multiple areas. In this discussion, we examine why the data sets from the same program led to two very different stories.

Our first interpretation of the differences between the two stories is that we documented a high degree of sharing between the study groups before the WLC group received the intervention, which is referred to as contamination in Western research. Prior to their taking part in the intervention, we asked the WLC participants if they had received information about the intervention and 80% (*n* = 74 out of 93) stated yes. The vast majority also provided details on actions they took to improve their health due to what they learned prior to their scheduled participation in the intervention. Sharing is a valuable part of community functioning within the Apsáalooke community and within other Indigenous communities [35]. Sharing can diminish comparative group findings in pragmatic clinical trials [36,37,38] and hence may have contributed to our non-significant quantitative findings. For more information on sharing within our intervention, see Allen et al. [39].

Our second interpretation of the differences between the two stories was that the quantitative measures we used were not culturally consonant for our participants. There were multiple indicators of this. As Walls and colleagues stated, “*Getting measurement right is essential. This is true for all research, but particularly so for Indigenous communities where rigorous measurement tools may not work as intended or have not been evaluated for cultural fit*” [40] (p. 12). Non-Indigenous partners selected the surveys, which were reviewed question-by-question in group meetings with Indigenous partners for use in the study. The understanding was that quantitative surveys would meet a need from the funding source and enhance the potential for the intervention to be deemed evidence-based by Western standards. Being evidence-based would allow the Apsáalooke and other communities the opportunity to receive funding to provide the program. Indigenous-developed self-management interventions did not exist when we created our program. Indigenous partners reviewed and accepted the surveys, and once data collection started, saw issues with the measures, which are discussed in more detail below. Furthermore, the process of gathering quantitative data (a participant answering questions on a computer tablet, versus two or more people having a conversation) was not relational, which would have been a culturally consonant method for gathering program impacts.

An indicator that the measures may not be culturally consonant is that most of our study’s quantitative measures used Likert response scales. Research has demonstrated that there are cultural differences in the selection of a Likert response option, with some refraining from selecting extreme response options (i.e., “strongly disagree” or “strongly agree”) [41]. This resonated with our Apsáalooke research partners, who shared that they were taught that it is good to be humble, not prideful, and to not take an extreme position on an issue. This questions the appropriateness of the Likert scales used in our study.

Another indicator that the measures may not be culturally consonant is that for the first three data collections, we asked participants the same questions for demographic data that were likely invariable (e.g., marital status and educational level). Some participants provided different responses for these data at different timepoints, and program staff verified with participants that the information had not changed. Additionally, based on unanticipated findings from baseline data and feedback from participants that one of the surveys was not clear, it was eliminated for future data collection events. Based on these findings from the demographic data and another survey, it is possible that other survey questions were not clear to the participants.

An additional indicator that the measures may not be culturally consonant was that some Aaakbaabaaniilea and staff (story data) received comments from some participants that this was the case. Examples included participants (1) having language and understanding barriers, (2) not relating to the questions, (3) skimming through the survey answering no to questions or responding to questions randomly without giving much thought to the answers, (4) feeling the survey was too long, and (5) being uncomfortable answering questions on a computer tablet. An example of questions not being relatable were items in several surveys that used the word confident, which was not familiar to some participants and is another example of a culturally inconsonant response (e.g., to state that one is “very confident”). An example of answering questions without paying much attention to content was a participant who was contacted for follow-up related to a high score on the PHQ-9 (a measure of depression). The respondent shared that she “just wanted to get it [the survey] done, I just answered it.” The above are all indicators that the measures may not have been appropriate in this community. One mentor received comments from a participant about the questions being good questions and one mentor thought that her participants were comfortable completing the surveys on a computer tablet. But the overwhelming evidence was that the measures may not have been culturally consonant.

The final indicator that the quantitative measures might not be culturally consonant for our population is that many Western-based measurements have not been fully validated in Indigenous communities and may impose beliefs and values inconsonant with these communities [42]. The importance of culturally valid Indigenous scientific measures and methods has been discussed by others [43]. In our study, the split half-reliability of the SF-12 subscales ranged from 0.37 to 0.72 in the current study, indicating low to moderate reliability, compared with 0.65 to 0.75 reported in the literature [44]. These multiple indicators lead us to believe that the quantitative measures were not appropriate measures to understand the impacts of Báa nnilah for the Apsáalooke Nation.

Our third interpretation of the differences between the two stories was that we did not include measures of changes in daily attitudes and behaviors in our pre–post quantitative surveys. We provided a list of these types of outcomes in a post-test-only measure and asked participants to check off changes they noticed for themselves resulting from participating in the program. Participants (*n* = 145) reported the following outcomes: eating healthier (64%), drinking more water (79%), having better communication with my doctor (60%), feeling more hopeful (58%), taking more walks or exercising more (57%), drinking less pop or other sugary drinks (57%), and having better relationships with my spouse or other family members (59%). We may have seen significant outcomes if we had included these types of measures in the pre–post quantitative assessment. As these measures were not a part of our hypotheses, we did not include them in the quantitative results above. However, they do provide evidence of program impacts. Additionally, the qualitative data were replete with evidence of changes in these types of outcomes.

Our fourth interpretation of the differences between the two stories is that participants who provided data via our three qualitative data sets experienced different outcomes from those who did not. Due to personal family obligations and family deaths, not every participant who was randomly selected to participate in qualitative interviews was able to participate (13 out of 20 participated). Additionally, while not every participant chose to enter responses regarding impacts into tablet computers or share impacts with program staff, 73% of participants entered comments into tablet computers at the fourth data collection and 93% entered comments at the fifth data collection. Finally, many participants shared stories of impactful outcomes with program staff. The similarity of findings across the three qualitative data sets and that most participants were represented in the qualitative data provide confidence that most participants had positive experiences. Participants had the opportunity to provide critical feedback through the qualitative interview and tablets and the only suggestion for improvement was to provide the program to more community members. We regard the possibility that the minority of participants without qualitative data may have had a less impactful experience. In sum, we believe that the lack of significance in the quantitative results was due to quantitative measurement issues. The qualitative assessments gave participants the opportunity to provide evidence of the program’s impact on themselves, their families, and the broader community.

There are multiple strengths related to our study outcomes. Strengths of program development and design are presented in other papers [3,4]. The primary strength is the noteworthy impact of the program that rippled out from participants to their families and the broader community that were assessed through qualitative methods. Outcomes included positive changes in self-management, diet, exercise, emotional health, relationships with healthcare providers, relationships with family members, and a broad sharing of program content. Many of the areas that showed substantive changes through the qualitative data were also measured with quantitative surveys, which did not show significant changes. Apsáalooke partners believe that it was the culturally consonant approach and content that caused changes, including the cultural strength of sharing good things with others and using a trauma-informed approach. The latter has been emphasized as important in interventions with Indigenous communities [45] and the use of this approach in our program is described by Schure et al. [4].

We also demonstrated the feasibility of recruiting and retaining community members in a seven-gathering CI self-management program. Assessments of disparities in participant retention in health interventions is a need expressed by others [46]. For participants with complete attendance data, nearly half attended all seven gatherings and over three-quarters attended five or more gatherings. This is especially meaningful considering that the community coped with challenges that may have impacted program attendance. Participants cited death in the family, personal or family crisis, severe weather, and personal health as the top barriers for program attendance. Finally, we established that the program is sustainable because participants were engaged in the program, shared what they learned, and are asking to participate again and asking for the program to be sustained in the community outside of a research study. This is noteworthy, as Western-based CI self-management programs have been shown to be impractical and difficult to sustain within Indigenous communities [47,48,49].

Our study has several limitations. As clinical measures were not collected, we could not assess the effects of the intervention on clinical outcomes. As mentioned above, we did not collect pre- and post-quantitative data on intermediate outcomes (e.g., exercising, drinking water), which may have shown significance. Also, our evaluation design was influenced by our striving to have Báa nnilah designated as an evidence-based program, which could make funds available for program sustainability, and by our hope for funding from the National Institutes of Health, who favor an RCT design for health intervention studies. Additionally, the time span between surveys for participants was large (1–10 months for time 1–2 [median = 5.5 months]; 2–8 months for time 2–3 [median = 5 months]). Allowing participants to complete the survey at their convenience increased our sample size.

The final limitation relates to our CBPR process used for selecting quantitative surveys, which was not led by Apsáalooke knowledge and understanding. As Walter and Anderson discussed, Indigenous knowledge, understanding, and knowers must “take the dominant position” [50] (p. 99) when seeking information from community members. Apsáalooke community partners review quantitative surveys that were selected by non-Indigenous university partners. To rectify this oversight, we are developing an Apsáalooke evaluation for the Báa nnilah program, with our CBPR process being led by Apsáalooke partners to develop meaningful outcomes, consonant with the culture.

## 5. Conclusions

A key factor in our evaluation design choice was our striving for program sustainability funding. In 2022, the US Health and Human Services Administration for Community Living (ACL) made USD 6 million available for communities to deliver and sustain evidence-based CI self-management programs. The programs used must be deemed evidence-based by ACL’s criteria, which utilize a Western research paradigm. This standard may result in minoritized communities applying for and delivering programs that are not consonant with their culture just to have something in their community, or communities deciding to not apply for funds because they only want to deliver programs that are consonant. ACL and other funds designated solely for programs that meet Western evidence-based criteria are examples of ongoing structural racism that causes further entrenchment of existing health disparities.

We strongly recommend that agencies who provide funding for evidence-based health programs using Western-based criteria expand their understandings of evidence. This includes community-defined and practice-based scientific evidence that has been utilized for time immemorial within Indigenous communities. Doing so honors and recognizes an Indigenous research paradigm and centers Indigenous voices and methods [39,51,52,53]. We also suggest that funding agencies promote and accept study designs that are more consonant with Indigenous cultures.

Our partnership is developing an Indigenous evaluation strategy to measure the impacts of the Báa nnilah program in a culturally consonant manner. The work is led by our Indigenous partners, including four Aaakbaabaaniilea, and is grounded in the values of respect, reciprocity, relationality, relevance, and responsibility [54,55]. Based on substantial program effects shown through qualitative data, we are sharing the Báa nnilah program with other Indigenous Nations. They are adapting the program manual to be consonant with their culture and we will evaluate outcomes using our Indigenous evaluation strategy. Finally, our team is consulting with national experts and funders regarding evidence-based designations to expand funding access for effective, culturally consonant programs. Our goal is to increase health and well-being across Indigenous communities.

## Figures and Tables

**Figure 1 ijerph-21-00285-f001:**
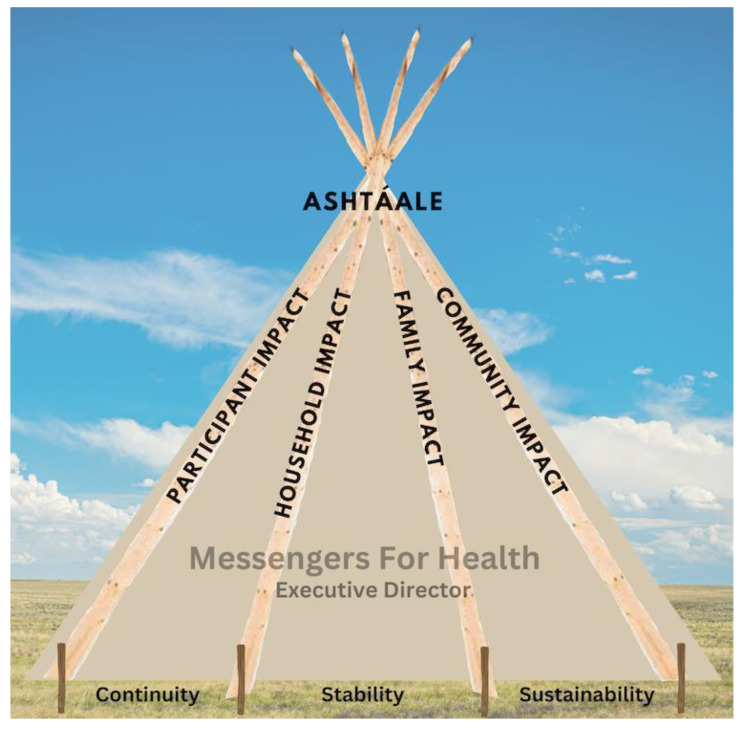
Ashtáaale (teepee) conceptual model.

**Table 1 ijerph-21-00285-t001:** Descriptive characteristics of the full sample and by study group.

	Full Sample *n* = 211		Intervention *n* = 104		Waitlist *n* = 107	
	*n*	*Mean ± SD* or	*n*	*Mean ± SD* or	*n*	*Mean ± SD* or
		*Frequency* (%)		*Frequency* (%)		*Frequency* (%)
Age at baseline	205	52.09 ± 13.47	101	51.71 ± 12.55	104	52.46 ± 14.36
Sex—Female	211	152 (72%)	104	69 (66%)	107	83 (78%)
Marital Status	211		104		107	
Married/In marriage-like relationship		102 (48%)		56 (54%)		46 (43%)
Separated or divorced		49 (23%)		18 (17%)		31 (29%)
Widowed		23 (11%)		12 (12%)		11 (10%)
Single, never married		36 (17%)		18 (17%)		18 (17%)
Other		1 (1%)		--		1 (1%)
Education	210		104		106	
Eighth grade or less		4 (2%)		4 (4%)		--
Some high-school		24 (11%)		11 (11%)		13 (12%)
High-school graduate or diploma		45 (21%)		23 (22%)		22 (21%)
At least some technical/vocational school or some college		66 (31%)		34 (33%)		32 (30%)
Associate’s degree		40 (19%)		16 (15%)		24 (23%)
Bachelor’s degree		18 (9%)		11 (11%)		7 (7%)
Post-graduate/professional degree		13 (6%)		5 (5%)		8 (8%)
Annual household income at baseline	208		104		104	
Under USD 10,000		68 (33%)		36 (35%)		32 (30%)
USD 10,000–USD 14,999		29 (14%)		16 (15%)		13 (13%)
USD 15,000–USD 24,999		41 (20%)		20 (19%)		21 (20%)
USD 25,000–USD 34,999		29 (14%)		16 (15%)		13 (13%)
USD 35,000–USD 49,999		22 (11%)		8 (8%)		14 (14%)
USD 50,000–USD 74,999		9 (4%)		3 (3%)		6 (6%)
USD 75,000–USD 99,999		7 (3%)		4 (4%)		3 (3%)
USD 100,000 and higher		3 (1%)		1 (1%)		2 (2%)
Insurance or health coverage	211		104		107	
Medicare/Medicaid		124 (59%)		60 (58%)		64 (60%)
Private Insurance		23 (11%)		17 (16%)		6 (6%)
Indian Health Service		95 (45%)		45 (43%)		50 (47%)
Food assistance programs	210		103		107	
Yes		116 (55%)		63 (61%)		53 (50%)
Self-reported ongoing illness(es)	211		104		107	
Diabetes		120 (57%)		68 (65%)		52 (49%)
Arthritis		71 (34%)		36 (35%)		35 (33%)
Heart disease		20 (10%)		12 (12%)		8 (8%)
Blood pressure		117 (56%)		61 (59%)		56 (52%)
Asthma/Lung disease/COPD		29 (14%)		16 (15%)		13 (12%)
Cancer		12 (6%)		7 (7%)		5 (5%)
Chronic pain		73 (35%)		34 (33%)		39 (36%)
Other		53 (25%)		27 (26%)		26 (24%)
Number of illness(es)	211	2.35 ± 1.25	104	2.51 ± 1.32	107	2.19 ± 1.17
Number of miles to the clinic	197	18.91 ± 19.63	96	20.93 ± 21.08	101	17.00 ± 18.04
Number of doctor visits in the past 4 months	197	4.13 ± 6.35	100	4.69 ± 8.18	97	3.55 ± 3.56
Number of people in the household	200	4.82 ± 2.65	97	5.09 ± 2.75	103	4.57 ± 2.53

**Table 2 ijerph-21-00285-t002:** Mean and standard deviation (SD) of outcome measures by study group at T1 and T2.

	Intervention		Waitlist Control	
Outcome Measures	T1 *Mean* ± *SD*	T2 *Mean* ± *SD*	T1 *Mean* ± *SD*	T2 *Mean* ± *SD*
Physical Test				
TUG Mean	20.83 ± 5.43	19.32 ± 5.07	20.39 ± 5.30	19.25 ± 4.27
Balance (mBESS Total)	13.41 ± 6.33	14.29 ± 7.24	13.81 ± 6.60	12.62 ± 6.57
6 min walk	1260.23 ± 287.70	1338.83 ± 276.47	1298.85 ± 265.40	1356.41 ± 319.59
Health Surveys				
SF12—Physical Health	44.99 ± 10.91	42.19 ± 11.35	45.05 ± 9.74	42.33 ± 9.99
SF12—Mental Health	47.06 ± 10.74	49.42 ± 11.02	46.58 ± 10.74	49.15 ± 10.8
Depression—PHQ-9	5.29 ± 5.88	5.44 ± 6.07	7.22 ± 7.47	6.21 ± 6.16
PROMIS—Satisfaction with Social Roles and Activities	53.62 ± 10.11	53.15 ± 10.87	54.03 ± 9.62	53.73 ± 9.57
PROMIS—Self-Efficacy for Managing Symptoms	49.33 ± 9.64	48.28 ± 9.41	48.15 ± 9.37	47.33 ± 8.36
PROMIS—Emotional Support	52.71 ± 9.65	53.20 ± 8.95	51.65 ± 11.09	53.45 ± 9.23
PROMIS—Positive Affect and Well-Being	56.32 ± 7.92	57.82 ± 8.22	56.37 ± 8.98	57.40 ± 8.37
PROMIS—Physical Function	44.94 ± 8.81	45.57 ± 9.09	45.49 ± 8.74	45.78 ± 8.53
PROMIS—Self-Efficacy for Managing Emotions	48.18 ± 9.13	49.80 ± 9.32	48.28 ± 9.84	49.38 ± 9.90
PROMIS—Self-Efficacy for Managing Social Interactions	46.57 ± 8.57	47.10 ± 8.59	47.44 ± 9.74	47.86 ± 8.94
Patient Activation Measure	69.31 ± 17.73	67.39 ± 21.09	65.5 ± 16.41	70.62 ± 19.83

**Table 3 ijerph-21-00285-t003:** Fixed effects, B (SE), of time-invariant and time-varying covariates based on HLM analyses.

	Group ^a^	Baseline Age	Sex ^b^	Educational Level ^c^				Number of Illnesses	Attendance	Intervention	Time from Baseline	Time by Group
				Some High-School	High-School Diploma	Some Technical School	At least some college					
TUG Mean	1.959 (0.911) *	0.166 (0.030) ***	−1.620 (0.825)	0.357 (1.264)	−0.967 (1.108)	−0.909 (1.173)	−2.304 (1.295)	0.084 (0.311)	−0.042 (0.195)	1.048 (0.582)	0.001 (0.001)	−0.006 (0.002) ***
mBESS Total	0.838 (1.439)	0.189 (0.038) ***	0.097 (1.047)	2.061 (1.597)	1.086 (1.417)	0.680 (1.477)	0.619 (1.653)	0.352 (0.397)	−0.248 (0.254)	2.004 (1.158)	0.011 (0.003) ***	0.009 (0.003) **
6 min walk	−86.753 (64.318)	−8.738 (2.054)	74.761 (57.127)	53.354 (89.058)	−11.997 (76.687)	−1.870 (81.099)	70.891 (89.007)	−6.441 (22.245)	−8.153 (13.582)	−23.006 (48.364)	−0.077 (0.108)	−0.168 (0.138)
SF-12 Physical	1.236 (1.415)	−0.099 (0.049) *	1.774 (1.370)	2.430 (2.120)	2.407 (1.942)	−0.366 (2.092)	3.539 (2.311)	−2.702 (0.514) ***	0.358 (0.341)	−0.147 (0.921)	0.005 (0.002) *	−0.002 (0.003)
SF-12 Mental	2.524 (1.620)	0.001 (0.054)	2.708 (1.501)	0.852 (2.336)	1.196 (2.134)	−0.584 (2.282)	2.172 (2.528)	−1.788 (0.574) **	0.656 (0.386)	0.421 (1.032)	0.007 (0.003) **	−0.004 (0.003)
PHQ-9	0.232 (1.005)	0.010 (0.036)	−0.793 (1.000)	−0.952 (1.568)	−1.246 (1.345)	1.519 (1.450)	−1.309 (1.571)	0.311 (0.375)	−0.444 (0.228)	0.124 (0.564)	0.002 (0.001)	0.001 (0.002)
Physical Function	−2.067 (1.298)	−0.198 (0.049) ***	2.912 (1.384) *	2.355 (2.174)	0.910 (1.851)	0.522 (2.010)	1.510 (2.157)	−2.486 (0.517) ***	0.452 (0.308)	0.452 (0.649)	0.001 (0.001)	0.002 (0.002)
SEMS	−0.322 (1.471)	−0.058 (0.051)	−0.494 (1.449)	1.810 (2.273)	1.678 (1.932)	0.074 (2.091)	2.811 (2.245)	−1.430 (0.540) **	−0.083 (0.321)	−0.398 (0.991)	0.002 (0.002)	0.001 (0.003)
EMSU	−0.422 (1.774)	0.007 (0.054)	−1.172 (1.459)	2.640 (2.269)	2.445 (2.007)	0.627 (2.112)	0.424 (2.386)	−0.134 (0.556)	0.222 (0.370)	0.494 (0.930)	0.002 (0.002)	0.001 (0.002)
SEME	−0.670 (1.591)	0.007 (0.055)	−0.094 (1.506)	2.211 (2.357)	1.223 (2.039)	0.894 (2.185)	1.562 (2.393)	−0.830 (0.568)	0.036 (0.354)	0.255 (0.868)	0.003 (0.002)	0.003 (0.002)
PAWB	−1.134 (1.449)	−0.013 (0.050)	0.001 (1.370)	2.439 (2.143)	1.527 (1.858)	0.715 (1.991)	2.120 (2.183)	−0.336 (0.517)	0.224 (0.323)	−0.033 (0.770)	0.001 (0.002)	0.001 (0.002)
SESI	−1.216 (1.515)	−0.003 (0.053)	−1.666 (1.469)	4.342 (2.302)	3.732 (1.977)	1.872 (2.128)	2.694 (2.310)	−0.272 (0.551)	−0.069 (0.337)	0.537 (0.899)	0.002 (0.002)	0.001 (0.002)
SATS	−1.782 (1.581)	0.007 (0.058)	0.026 (1.617)	2.957 (2.537)	0.105 (2.167)	−0.614 (2.345)	−1.443 (2.527)	−1.766 (0.605) **	0.371 (0.363)	0.478 (0.873)	0.001 (0.002)	0.001 (0.002)
PAM	−1.621 (3.102)	−0.179 (0.097)	−2.907 (2.723)	4.237 (4.266)	9.071 (3.649) *	3.653 (3.919)	10.284 (4.257) *	0.590 (1.016)	1.145 (0.627)	1.994 (2.345)	0.014 (0.007) *	−0.009 (0.007)

Note: * *p* < 0.05; ** *p* < 0.01; *** *p* < 0.001; ^a^: the referent category is the wait list control group; ^b^: the referent category is female; ^c^ the referent category is 8th grade or lower.

## Data Availability

Data from this study are unavailable due to privacy restrictions.

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
