# Peer review of "The Báa nnilah Program: Results of a Chronic-Illness Self-Management Cluster Randomized Trial with the Apsáalooke Nation"

_ijerph, 2024, doi:10.3390/ijerph21030285_

Round 1
Reviewer 1 Report
Comments and Suggestions for Authors
Thank you for submitting this very thoughtful and culturally sensitive research study on a chronic illness self-management program in an indigenous community. The mixed methods design and the community based participatory research approach generated powerful qualitative data and findings that challenged the use of standardized Western research instruments to measure concepts that are culturally grounded. The manuscript is very thorough in its analysis of findings and implications for future study in this area. I also appreciate how the authors attempted to address the long-term impacts of the program by measuring outcomes over defined time periods. I was a little unclear as to why pre-baseline data was obtained in the waitlist control group. What is the time frame difference between the pre-baseline and baseline period? What was the rationale for collecting data pre-baseline on the waitllst control group only? Also, the measure for patient activation seems to overlap with the concept of self-efficacy which was also measured by another scale. Was there an inter-correlation between these two measures? Was the self-confidence component of the patient activation scale essentially a measure of self-efficacy? In terms of the instruments to measure physical functioning, why was the distance walked changed from 3 to 10 meters for this population (p. 4, Line 152) and how was change in number of meters determined? Why was balance assessment modified? (Lines 156-157). Do you have any thoughts around why the intervention group scored lower on physical well-being and depression and had more errors in static balance than the control group? Finally, have you considered adding a visual diagram to present your conceptual model that arose from the qualitative data? I believe it would be visually compelling and assist the reader to grasp the dimensionality and relational connections between the concepts. Thank you so much for contributing this significant addition to the research literature.
I might suggest is to remove some of the qualitative examples as the manuscript is rather lengthy and I do believe 2 or 3 good narrative examples would suffice in demonstrating how the conceptual model surfaced from the data. Also, a visual diagram or picture of the conceptual model of the teepee with canvas, stakes, etc. that signify the relational changes and the impact on the individual and family and community would provide a visual representation of the conceptual model supported by this data.
Comments on the Quality of English LanguageThe manuscript, overall, reads smoothly and fluently.
Author Response
Reviewer 1:
Comment: Thank you for submitting this very thoughtful and culturally sensitive research study on a chronic illness self-management program in an indigenous community. The mixed methods design and the community based participatory research approach generated powerful qualitative data and findings that challenged the use of standardized Western research instruments to measure concepts that are culturally grounded. The manuscript is very thorough in its analysis of findings and implications for future study in this area. I also appreciate how the authors attempted to address the long-term impacts of the program by measuring outcomes over defined time periods.
Response: Thank you for your kind comments and for reviewing and providing suggestions to make our manuscript stronger. We appreciate your time and thoughtfulness. We have taken the time to carefully review and respond to your review. What follows is each of your comments followed by a response from our team including where and how the revised manuscript has been changed. The changes are highlighted in yellow in the revised manuscript.
Comment: I was a little unclear as to why pre-baseline data was obtained in the waitlist control group. What is the time frame difference between the pre-baseline and baseline period? What was the rationale for collecting data pre-baseline on the waitllst control group only?
Response: Typically, with a classical experimental design, both the Intervention and the Control groups are measured at the baseline (T1), then, the intervention is implemented among the Intervention Group only, followed by both groups being measured again (T2). Typically, T1 is called “baseline” (or “pretest”) and T2 is called “post-intervention” (or “posttest”). Our longitudinal research design included all the above components of a classical experimental design. In addition, we measured both study groups after the Waitlist Control Group had received the intervention (T3&T4), and followed-up with the Waitlist Control Group one more time (T5) so that they were followed up for the same length of time post-intervention as the Intervention group. In order to label each time of measurement in relation to each study group’s exposure to the intervention, we called T1 “Baseline” for the Intervention Group and “Pre-baseline” for the Waitlist Control Group, although they refer to the same time historically speaking. In sum, this is purely semantic, and did not affect the analysis. Data collected from all participants in both study groups at all waves of measurements were included in the quantitative analyses. We added T1-T5 on lines 108-111 to provide clarification.
Comment: Also, the measure for patient activation seems to overlap with the concept of self-efficacy which was also measured by another scale. Was there an inter-correlation between these two measures? Was the self-confidence component of the patient activation scale essentially a measure of self-efficacy?
Response: In our study, the measure of patient activation was significantly correlated with each of the measures of Self-Efficacy for Managing Symptoms (r = .377, p < .01), Self-Efficacy for Managing Emotions (r = .416, p < .01), and Self-Efficacy for Managing Social Interactions (r = .416, p < .01). These bivariate correlations are in the low-to-moderate range, indicating that the patient activation measure is not essentially a measure of self-efficacy. We are not sure of the meaning of the reviewer’s comment “self-confidence component of the patient activation scale.” There are no separate components (subscales) for this measure and the scoring of this survey was conducted per the scoring instructions by its creators. The Patient Activation Measure has documented reliability and validity and the papers regarding this is in our manuscript as references 23 and 24. We did not make a change in the manuscript regarding this comment.
Comment: In terms of the instruments to measure physical functioning, why was the distance walked changed from 3 to 10 meters for this population (p. 4, Line 152) and how was change in number of meters determined?
Response: The distance walked was changed from 3 to 10 meters due to an error from a consultant who relayed the incorrect distance for this measure to our research group. We added this into the manuscript on lines 154-155.
Comment: Why was balance assessment modified? (Lines 156-157).
Response: We decided to alter the test to make it more possible for participants to complete as they had chronic health conditions and this test proved difficult for them when we piloted it. We added this information to the manuscript on lines 161-163.
Comment: Do you have any thoughts around why the intervention group scored lower on physical well-being and depression and had more errors in static balance than the control group?
Response: We discussed this question in our writing team. Our guess is that it is a random outcome due to factors that we did not measure such as traumas, for example, deaths among certain families. We did not make changes to the manuscript regarding this comment.
Comment: Finally, have you considered adding a visual diagram to present your conceptual model that arose from the qualitative data? I believe it would be visually compelling and assist the reader to grasp the dimensionality and relational connections between the concepts. Thank you so much for contributing this significant addition to the research literature.
Response: We developed a visual of our conceptual model and it is described and included at lines 334-337.
Comment: I might suggest is to remove some of the qualitative examples as the manuscript is rather lengthy and I do believe 2 or 3 good narrative examples would suffice in demonstrating how the conceptual model surfaced from the data. Also, a visual diagram or picture of the conceptual model of the teepee with canvas, stakes, etc. that signify the relational changes and the impact on the individual and family and community would provide a visual representation of the conceptual model supported by this data.
Response: Our writing team carefully considered the comment suggesting that we remove some of the qualitative examples. We would like to keep each example in the manuscript as we feel that each story that is presented is important in showing the deep and significant impact of the program. Stories are highly respected and valued in the Apsáalooke culture and are seen as a powerful way to honor the story teller and their experience. We wrote about the power and use of stories in another article (reference 7 in our reference list). We did not make changes to the manuscript regarding this comment.
We developed a visual of our conceptual model and it is described and included at lines 334-337.
Reviewer 2 Report
Comments and Suggestions for Authors
Thank you for the opportunity to review this paper. Firstly, I would like to congratulate the authors and the study team for undertaking this work - it is incredibly important and to see the efforts the team went to in order to deliver a culturally appropriate intervention is encouraging.
Overall the paper was very well written and I have only minor comments and wish to seek a few points of clarification.
Methods:
line 71 - what was your primary endpoint?
line 102 - what was the actual effect size the study was powered to detect? You have noted this for the interaction but not for the primary outcome.
line 110 - I was intrigued as to why there was a pre-baseline measurement for the WLC group but not the intervention group. Could you provide some additional context on this and advise whether this was factored in to analysis at all?
line 177 - It may just be my lack of familiarity with the statistical approach, but I struggled to conceptualise the "Intervention" co-variate and how this was applied if a measure was collected multiple times post intervention. For example, if you had a mean TUG score at baseline, post intervention, 6 and 12 mth post intervention, which of these post intervention scores is used as the outcome? While it makes sense to adjust for the baseline score as a co-variate (and any previous post intervention scores for outcomes collected later), I would expect to see a beta score for each follow-up time period. Based on my reading of your methods, it is this "intervention" variable that is telling you if you had a main effect of the intervention (compared to control) but I don't follow this given "intervention" is really a proxy of time? This approach is difficult to follow and requires additional explanation.
line 281 - the Beta values reported for the interactions for TUG mean and mBESS total dont appear to match the values in Table 3.
line 296 - you state here that mBESS total scores significantly improved over time, which seems at odds with your finding reported at line 285 of mBESS scores worsening over time. Given the effect of time seems to be driving this interaction (Table 3) rather than the effect of group is one of these conclusions incorrect? These 2 results dont seem like they can both be right
Results general - I wonder whether there would be some benefit to including the mean (SD) scores for the outcome measures by study group, for post-intervention at the very least? It would provide additional context for the reader and may demonstrate clinical significant changes in some measures, even if statistical significance was not reached.
Discussion general - while I agree that the quantitative measures may not have been culturally consonant, and this may be one reason for the disparate effects shown in the qualitative and quantitative results, this would be presumably be less of an issue for the physical tests (which interestingly, is where you found your effects). I also wonder if you had used measures more proximal to the intervention focus (such as some sort of food diary/self reported intake of sugar sweetened beverages, pedometers to measure steps taken) or other physical measures (such as blood pressure, BMI) whether you may have also demonstrated effects. Obviously finding measures that are acceptable to your target population is critical and this would need to be determined in conjunction with your cultural advisory group. Perhaps something that could be considered for future work?
Thank for your work in this area.
Author Response
Reviewer 2:
Comment: Thank you for the opportunity to review this paper. Firstly, I would like to congratulate the authors and the study team for undertaking this work - it is incredibly important and to see the efforts the team went to in order to deliver a culturally appropriate intervention is encouraging.
Overall the paper was very well written and I have only minor comments and wish to seek a few points of clarification.
Response: Thank you for your kind comments and for reviewing and providing suggestions to make our manuscript stronger. We appreciate your time and thoughtfulness. We have taken the time to carefully review and respond to your review. What follows is each of your comments followed by a response from our team including where and how the revised manuscript has been changed. The changes are highlighted in yellow in the revised manuscript.
Comment: line 71 - what was your primary endpoint?
Response: The primary endpoint for a classic experimental design would be T2 (post-test). However, in our longitudinal study, we followed the participants at 6-months and 12-months post-intervention to evaluate the effects of participating in the program over a longer time period. We did not make a change in the manuscript regarding this comment.
Comment: line 102 - what was the actual effect size the study was powered to detect? You have noted this for the interaction but not for the primary outcome.
Response: The interaction effect (which tested the effect of the intervention) was for the primary outcome. We did not make a change in the manuscript regarding this comment.
Comment: line 110 - I was intrigued as to why there was a pre-baseline measurement for the WLC group but not the intervention group. Could you provide some additional context on this and advise whether this was factored in to analysis at all?
Response: This comment was also provided by Reviewer 1 and our response is copied from our response to that reviewer. Typically, with a classical experimental design, both the Intervention and the Control groups are measured at the baseline (T1), then, the intervention is implemented among the Intervention Group only, followed by both groups being measured again (T2). Typically, T1 is called “baseline” (or “pretest”) and T2 is called “post-intervention” (or “posttest”). Our longitudinal research design included all the above components of a classical experimental design. In addition, we measured both study groups after the Waitlist Control Group had received the intervention (T3&T4), and followed-up with the Waitlist Control Group one more time (T5) so that they were followed up for the same length of time post-intervention as the Intervention group. In order to label each time of measurement in relation to each study group’s exposure to the intervention, we called T1 “Baseline” for the Intervention Group and “Pre-baseline” for the Waitlist Control Group, although they refer to the same time historically speaking. In sum, this is purely semantic, and did not affect the analysis. Data collected from all participants in both study groups at all waves of measurements were included in the quantitative analyses. We added T1-T5 on lines 108-111 to provide clarification.
Comment: line 177 - It may just be my lack of familiarity with the statistical approach, but I struggled to conceptualise the "Intervention" co-variate and how this was applied if a measure was collected multiple times post intervention. For example, if you had a mean TUG score at baseline, post intervention, 6 and 12 mth post intervention, which of these post intervention scores is used as the outcome? While it makes sense to adjust for the baseline score as a co-variate (and any previous post intervention scores for outcomes collected later), I would expect to see a beta score for each follow-up time period. Based on my reading of your methods, it is this "intervention" variable that is telling you if you had a main effect of the intervention (compared to control) but I don't follow this given "intervention" is really a proxy of time? This approach is difficult to follow and requires additional explanation.
Response: The statistical approach that we used, hierarchical linear modeling, was a longitudinal and multilevel analysis. With the example used by Reviewer 2 above, the TUG mean score at each time of measurement is the outcome measure in this longitudinal study. At Level 1 (the individual level) of the longitudinal analysis, each individual participant’s mean TUG score at baseline, post intervention, 6 and 12 month post intervention were used to obtain a best-fitting line (using the least square criterion) using the linear function. This best fitting line (also called the growth curve) allows us to examine each participant’s intraindividual change over time. Its intercept represents the baseline measure, and its slope represents the rate of change in the outcome measure over time. Thus, there is not a beta score for each follow-up time point, but one beta score (i.e., the slope of the best fitting line) which indicates how fast each individual’s mean TUG score changed over time. The dummy-coded "Intervention" variable (1= participated in the intervention; 0 = did not participate in the intervention) is another time-varying co-variate, which was included in the analytical model because the change of the mean TUG score could be caused by a time effect and by whether each individual participated in the intervention or not at each time of measurement. Including the "Intervention" co-variate as a Level-1 predictor in addition to “time” allows us to separate the Time and the Intervention effects statistically.
At Level 2 of the longitudinal analysis, the individual best fitting lines obtained at Level 1 were examined for interindividual differences in change (e.g., which independent variables at Level 2 are significant predictors of the rate of change in the mean TUG score). We did not make a change in the manuscript regarding this comment.
Comment: line 281 - the Beta values reported for the interactions for TUG mean and mBESS total don’t appear to match the values in Table 3.
Response: The difference was due to rounding and showing 2 versus 3 digits after the decimal. We changed the values in the text to match the values in the table. You can find this on line 286.
Comment: line 296 - you state here that mBESS total scores significantly improved over time, which seems at odds with your finding reported at line 285 of mBESS scores worsening over time. Given the effect of time seems to be driving this interaction (Table 3) rather than the effect of group is one of these conclusions incorrect? These 2 results don’t seem like they can both be right.
Response: Lower mBESS scores indicate better balance. Thus, a decrease in the mBESS total score indicates improvement over time. We added this information in the methods section on line 158.
Comment: Results general - I wonder whether there would be some benefit to including the mean (SD) scores for the outcome measures by study group, for post-intervention at the very least? It would provide additional context for the reader and may demonstrate clinical significant changes in some measures, even if statistical significance was not reached.
Response: We added two columns to Table 2 showing the means (SD) for post-intervention. You can find the revised table on line 295.
Comment: Discussion general - while I agree that the quantitative measures may not have been culturally consonant, and this may be one reason for the disparate effects shown in the qualitative and quantitative results, this would be presumably be less of an issue for the physical tests (which interestingly, is where you found your effects). I also wonder if you had used measures more proximal to the intervention focus (such as some sort of food diary/self reported intake of sugar sweetened beverages, pedometers to measure steps taken) or other physical measures (such as blood pressure, BMI) whether you may have also demonstrated effects. Obviously finding measures that are acceptable to your target population is critical and this would need to be determined in conjunction with your cultural advisory group. Perhaps something that could be considered for future work?
Response: We agree that it would have been beneficial to include pre-post measures that were more proximal to the intervention focus. On lines 569-570 we share that “We may have seen significant outcomes if we had included these types of measures in the pre-post quantitative assessment” and on lines 618-621 we bring this up as a limitation of our study. We agree that including more proximal measures is something for us to consider for future work.